# Medical Report Generation via Multimodal Spatio-Temporal Fusion

## ABSTRACT

Medical report generation aims at automating the synthesis of accurate and comprehensive diagnostic reports from radiological images. The task can significantly enhance clinical decision-making and alleviate the workload on radiologists. Existing works normally generate reports from single chest radiographs, although historical examination data also serve as crucial references for radiologists in real-world clinical settings. To address this constraint, we introduce a novel framework that mimics the workflow of radiologists. This framework compares past and present patient images to monitor disease progression and incorporates prior diagnostic reports as references for generating current personalized reports. We tackle the textual diversity challenge in cross-modal tasks by promoting style-agnostic discrete report representation learning and token generation. Furthermore, we propose a novel spatio-temporal fusion method with multi-granularities to fuse textual and visual features by disentangling the differences between current and historical data. We also tackle token generation biases, which arise from long-tail frequency distributions, proposing a novel feature normalization technique. This technique ensures unbiased generation for tokens, whether they are frequent or infrequent, enabling the robustness of report generation for rare diseases. Experimental results on the two public datasets demonstrate that our proposed model outperforms state-of-the-art baselines.

## KEYWORDS

Medical report generation, Multimodal Fusion, Cross-modal generation

**ACM Reference Format:**
. 2018. Medical Report Generation via Multimodal Spatio-Temporal Fusion. In *Proceedings of Make sure to enter the correct conference title from your rights confirmation emai (Conference acronym 'XX)*. ACM, New York, NY, USA, 10 pages. https://doi.org/XXXXXXX.XXXXXXX

## 1 INTRODUCTION

Radiological imaging is crucial for medical diagnosis, with the resultant reports being essential for clinical decision-making. However, the increasing demand for these services has significantly burdened radiologists, particularly affecting report quality and increasing the potential for errors [8, 18, 36]. This challenge necessitates the exploration of automatic radiology report generation systems. Current research efforts focus on automating the generation of reports, and enhancing the efficiency and quality of generated reports via the utilization of multimodal processing methods rooted in the computer vision and natural language processing domains.

Recent works [4, 24–26, 29] for medical report generation leveraged medical tags [16, 46, 47], pretrained models [1, 32], and cross-modal memory networks [6, 35] to enhance the clinical accuracy and quality of the generated reports. Despite these advancements, existing methods mainly produce reports from individual chest radiographs, overlooking the complexity of disease progression and the valuable insights provided by historical reports for current report generation. In clinical practice, handling follow-up patients involves integrating data from past examinations into new reports. To address this, some researchers [3, 33] combined both previous and current medical images to formulate reports. However, approaches that solely rely on tracing historical visual features overlook the significance of patients' past diagnostic reports in textual form. In practice, radiologists analyze both previous and current images to assess disease progression, enhancing earlier reports with updated descriptions of the disease's evolution to compile comprehensive current reports. Referring to prior reports helps doctors compose coherent and consistent report content across different diagnostic instances.

In this paper, we introduce a novel report generation framework designed to align with the workflow of radiologists. This framework compares previous and current images of patients to identify the disease progression, incorporating previous reports and simulating their writing style to generate current ones. Given the challenges presented by textual diversity in cross-modal generation tasks, we propose an enhanced diagnostic report generation method via learning style-agnostic discrete representations of reports and predicting tokens accordingly. To achieve high-quality discrete representations of reports, we have developed the RadFusion module, which conducts multi-granular spatio-temporal fusion of textual and visual features within the patient's clinical context. Additionally, we have developed a novel feature normalization technique to address the challenges posed by the long-tail distribution of token frequencies in current report generation. This technique employs linear projection to adjust the initial semantic features of tokens, ensuring that their utilization is not biased by token frequencies, thus, mitigating the discrepancy in prediction likelihood between high-frequency and low-frequency tokens.

Our proposed method is evaluated on two public datasets, i.e., MIMIC-CXR [17] and MIMIC-ABN [31]. The experimental results demonstrate the improvements of our method in both language quality (ranking the highest across all six natural language generation evaluation metrics) and factual statement accuracy (+ 3% F1-RadGraph and + 1.8% in F1-Chexbert on MIMIC-CXR) over strong baselines, including large language models (LLMs). More importantly, our proposed feature normalization method achieves higher performance gains on infrequent disease types (+ 8.6% F1 on MIMIC-ABN) than on frequent ones (+ 4.7% F1 on MIMIC-ABN). The ethical implications of this improvement are substantial within the clinical domain. While traditional machine learning excels at identifying

common patterns, its ability to generalize to less common patterns is relatively limited [28]. In clinical contexts, overlooking uncommon cases to achieve higher accuracy is not feasible. Our feature normalization method addresses this issue by refining wording preferences, thereby enhancing the accuracy and objectivity of factual descriptions of non-common diseases.

In summary, this paper makes the following contributions: (1) We propose a new framework that emulates radiologists' review processes, comparing historical and current images to detect disease progression and integrating previous reports to generate current reports. (2) We develop the RadFusion module to facilitate multi-granular spatio-temporal fusion of textual and visual features within a patient's clinical context. (3) We develop a feature normalization technique to tackle the long-tail distribution challenge in token frequency, improving factual statement accuracy across common and non-common diseases.

## 2 RELATED WORKS

In recent times, artificial intelligence has seen extensive utilization within the medical field [5, 9, 27, 43]. As a task that generates text from images, most medical report generation methods have adopted the encoder-decoder framework popularized in image captioning tasks. Initial efforts [37] utilized an encoder-decoder framework that combines Convolutional Neural Networks (CNN) for image encoding with Recurrent Neural Networks (RNN) for text decoding. Unlike concise image captions, medical reports entail elaborate long texts describing multiple organs and regions. To address this, Jing et al. [47] enhanced the CNN-RNN architecture by incorporating a co-attention module that merges visual and semantic features using disease tags, coupled with a hierarchical LSTM for crafting detailed report paragraphs. Additionally, recognizing the prevalence of normal samples over abnormal ones in medical reports, researchers have focused on mitigating data bias. CMAS-RL [15] refined the textual decoder through a multi-agent system, trying to balance descriptions of both normal and abnormal findings. Contrastive Attention [25] aimed to accentuate critical abnormalities by comparing the subject image against a normal image corpus. HRGR-Agent [22] merged retrieval-based and generative approaches for managing frequently normal and infrequently abnormal sentences.

Given the outstanding performance of pretrained models across various domains, recent works [1, 32] explored fine-tuning pre-trained visual encoders and textual decoders for medical report generation. Several researchers leveraged auxiliary signals to guide the generation of medical reports. Li et al. [19] extracted normal and abnormal terms from the MIMIC-CXR dataset to serve as nodes, with edges defined by attention weights between them, thus constructing a knowledge graph. This knowledge graph has been utilized by other researchers [21, 49] as a form of prior domain knowledge to enhance report generation. Additionally, relevant research [24, 48] developed heterogeneous graphs by associating 8 organs with 20 findings, where findings linked to the same organ are interconnected. Liu et al. [24] leveraged global representations derived from pre-retrieved reports within the training corpus to encapsulate domain-specific knowledge. Li et al. [20] dynamically updated the pre-constructed graph to model domain knowledge. In contrast to methods focusing solely on abnormalities, Jain et al. [14]

employed natural language processing tools to extract clinical entity and relation annotations from reports, thereby establishing the comprehensive radiological knowledge graph, RadGraph. Yang et al. [45] introduced general domain knowledge by learning the universal representations of the pre-constructed RadGraph. Other studies have concentrated on enhancing report generation through cross-modal alignment. Chen et al. [6] introduced the Cross-Modal Memory Network (CMM), which employs a shared memory for aligning images and texts, thereby enriching the quality of generated reports. Qin et al. [35] further advanced CMM by integrating reinforcement learning, employing natural language generation metrics as rewards to refine the cross-modal mappings for better image-text alignment. However, these methods treat image-report pairs within datasets as isolated from each other, disregarding the fact that radiologists frequently refer to comparisons with patients' previous examinations in their reports. Addressing this gap, Bannur et al. [3] aim to capture historical relevance by comparing patients' previous and current images, facilitating improved cross-modal alignment. Similarly, CXRmate [33] generates reports by integrating patients' previous reports with current images. Recap [10] also contrasts patients' prior and current images to deduce disease progression, thereby enhancing report generation.

To sum up, despite the progress made in medical report generation, there are several limitations that need to be addressed. (1) Radiologists' varied writing styles pose challenges to cross-modal alignment and the generation of disease-relevant content. (2) Historical reports of follow-up patients, which provide critical references for current clinical assessments of disease progression, were not integrated into the current report generation process, despite their importance in real-world practice. (3) Severe data imbalances complicate the detection and description of non-common diseases, leading to diagnostic biases in the learning and inference processes of machine learning models.

## 3 METHODOLOGY

We propose a spatio-temporal multimodal fusion method (see Figure 1a) to learn two tasks: inclusive node prediction, and radiological report generation. Task 1 learns to identify nodes that should be included in a current report, where the nodes are entities related to different radiographic observations. Task 2 aims to generate reports from the identified nodes. For a follow-up patient, the original input includes the historical examination data (both radiographs and reports) and current radiographs. For those without historical records, the original input is the current radiographs[1]. To generate style-agnostic discrete reports, the input also includes a global radiology knowledge graph for both types of patients.

The global radiology knowledge graph is developed from the training reports. The knowledge graph has nodes representing anatomical or observational entities and directed edges indicating the relationship between nodes. The prior report of a follow-up patient will be also converted into the sub-graph of the global graph. Converting textual reports into graphs (style-agnostic discrete representations) offers the advantage of filtering out stylistic wording variations present in the original contexts from various radiologists.

---

[1] In Figure 1a, the input does not contain the prior X-ray, the prior report and its associated nodes for new patients.

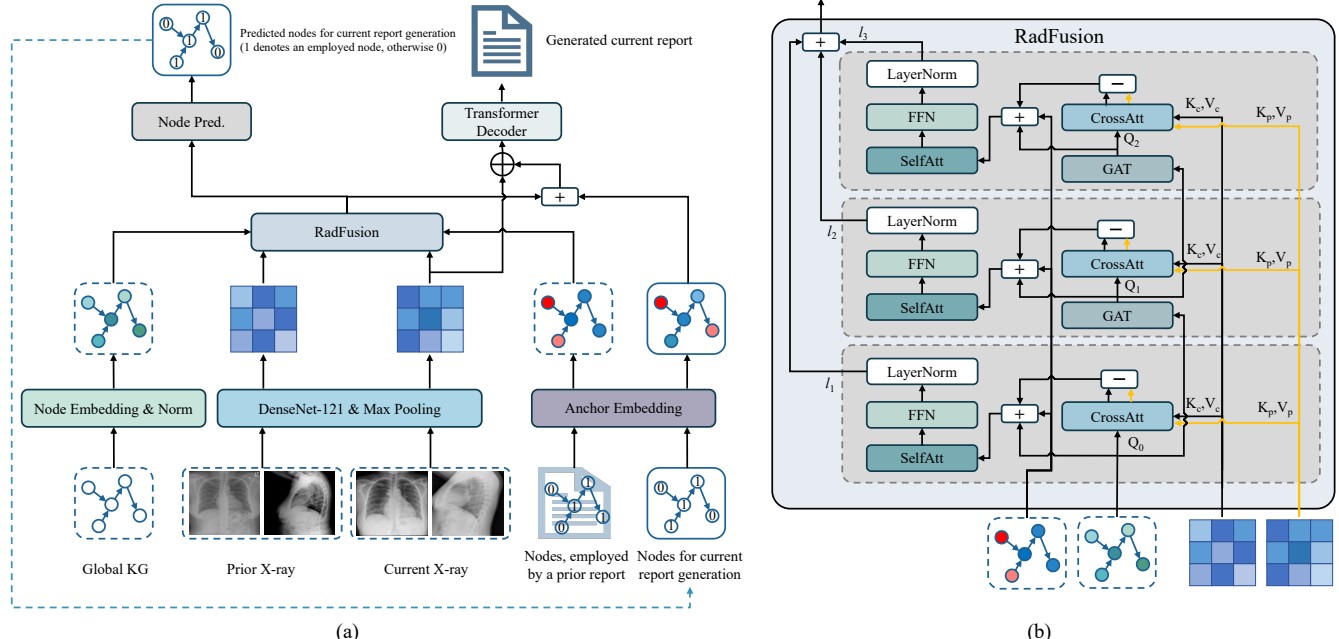

(a)         (b)

Figure 1: (a) The overall framework of our proposed model. (b) The proposed RadFusion module. ⊞ denotes matrix addition; ⊟ denotes subtraction; ⊕ denotes concatenation. Colored rounded rectangles denote computational layers with learnable parameters, while the white ones do not have learnable parameters. The graphs and matrices are the input or output of a computational layer.

These variations can otherwise disrupt the learning of modal alignment between factual information (such as entities) in the text and observations in radiographs.

To mitigate the biased impact of the uneven distribution of node frequencies, i.e., long-tail distribution, on the inclusive node prediction task (Task 1), we introduce a mathematics-explainable feature normalization method that operates on the node feature representations. This method projects the node features into a designated space (see Figure 2), rendering the prediction of nodes insensitive to their frequency attributes. Given input with spatial (e.g., the knowledge graphs) and temporal (e.g., the current and prior information) relationships, we also develop a RadFusion module (see Figure 1b) for the spatio-temporal fusion of the multimodal features (e.g., graphs and images). Ultimately, a comprehensive diagnostic report (Task 2) is generated by synthesizing the knowledge graph, enriched with current chest radiograph features, spatio-temporal and multimodal features, and the predicted discrete nodes.

## 3.1 Knowledge Graph Initialization

### 3.1.1 Construction.
To extract diagnostic visual features that are prioritized by radiologists, we propose harnessing radiological knowledge graphs to guide the cross-modal feature alignment and facilitate multimodal feature fusion. Utilizing the tool developed by Wang et al. [14], we structure entities mentioned in the reports in the training set into corresponding knowledge graphs. The nodes of these graphs represent either anatomical entities (e.g., lung, mediastinum) or observational entities (e.g., pneumonia). Edges are

directed and heterogeneous, capturing three types of relationships among entities: modify, located at, and suggestive of. The graphs from individual reports are amalgamated to form a comprehensive global knowledge graph $G$. Given the multiplicity of potential relationships between the same pair of entities and the relatively low semantic differentiation among the three types of edge relationships, we opt to disregard edge-type attributes within the global knowledge graph.

### 3.1.2 Node Embedding and Normalization.
We plan to identify diagnostic visual features associated with nodes from knowledge graph and predict their inclusion in the report. However, the nodes' frequency distribution reveals a long-tail curve, marked by prevalent common nodes (such as "lung", and "heart") versus infrequent abnormal findings. This distribution inherently biases node prediction towards frequently occurring nodes being identified in the report, while rarer findings are frequently overlooked. As illustrated in Figure 2, in predicting labels for the inclusion of nodes, we establish two anchor points, e.g., $S_0$ and $S_1$, representing the binary node labels of presence (1) and absence (0) within reports. High-frequency nodes tend to gravitate towards the vector space, associated with label 1, whereas low-frequency nodes are closer to label 0. This phenomenon imparts an inherent bias in the semantic features of nodes towards label prediction.

To mitigate this issue, we propose a method to normalize the node feature representations, eliminating innate label bias in node inclusion prediction while maintaining semantic differentiation. As

depicted in Figure 2, an equidistant hyperplane is initially determined based on anchor points $S_0$ and $S_1$, such that every point on this plane is equidistant to both nodes $S_0$ and $S_1$. Subsequently, a linear projection transformation is applied to map the initial node features onto this equidistant hyperplane. This process ensures that the updated node features $\mathbf{E}'^T$ are unbiased toward any label while preserving their semantic relationships in the vector space. The specific process is outlined as follows:

**Step1: Node and Anchor Embedding.** Node features $\mathbf{E}^G \in \mathfrak{R}^{M \times d}$ ($M$ denotes the number of nodes; $d$ denotes the dimension of embeddings.) within the constructed knowledge graph were initially randomized. Similarly, the features of the two anchor points $S_0$ and $S_1$ are defined as $\mathbf{s}_0$ and $\mathbf{s}_1$, respectively, both initialized randomly. Both node features and anchor point features are learnable.

**Step2: Equidistant hyperplane.** For any point $X$ on the equidistant hyperplane with coordinates $\mathbf{x}$, the equation of the equidistant hyperplane can be derived as follows:

$$|\mathbf{x} - \mathbf{s}_0|^2 = |\mathbf{x} - \mathbf{s}_1|^2 \tag{1}$$

Expanding the squared distances, we have

$$(\mathbf{x} - \mathbf{s}_0) \cdot (\mathbf{x} - \mathbf{s}_0) = (\mathbf{x} - \mathbf{s}_1) \cdot (\mathbf{x} - \mathbf{s}_1). \tag{2}$$

Simplifying Eq. 2 by expanding and rearranging terms yields

$$2(\mathbf{b} - \mathbf{s}_0) \cdot \mathbf{x} = |\mathbf{s}_1|^2 - |\mathbf{s}_0|^2. \tag{3}$$

**Step3: Equation of the Perpendicular.** For an initial node $E$, represented by coordinates $\mathbf{e}$, the aim is to project it onto the equidistant hyperplane, ensuring orthogonality between $E$ and its projection $E'$. Using the normal vector $\mathbf{n} = \mathbf{s}_1 - \mathbf{s}_0$ of the hyperplane, defined by vectors $\mathbf{s}_0$ and $\mathbf{s}_1$ of anchor points $S_0$ and $S_1$, we delineate the perpendicular from $E$ as a linear trajectory guided by $\mathbf{n}$.

$$\mathbf{x} = \mathbf{e} + t\mathbf{n}, \tag{4}$$

where $t$ symbolizes a scalar parameter indicative of the displacement along the direction vector $\mathbf{n}$.

**Step4: Target Mapping node $E'$.** To solve for the mapping point $E'$ and its coordinates $\mathbf{e}'$, substitute the equation of the perpendicular into the equation defining the equidistant hyperplane, thereby solving for the scalar parameter $t$.

$$2(\mathbf{s}_1 - \mathbf{s}_0) \cdot (\mathbf{e} + t(\mathbf{s}_1 - \mathbf{s}_0)) = |\mathbf{s}_1|^2 - |\mathbf{s}_0|^2 \tag{5}$$

Next, expanding this and solving for $t$ yields

$$2(\mathbf{s}_1 - \mathbf{s}_0) \cdot \mathbf{e} + 2t(\mathbf{s}_1 - \mathbf{s}_0) \cdot (\mathbf{s}_1 - \mathbf{s}_0) = |\mathbf{s}_1|^2 - |\mathbf{s}_0|^2, \tag{6}$$

$$t = \frac{|\mathbf{s}_1|^2 - |\mathbf{s}_0|^2 - 2(\mathbf{s}_1 - \mathbf{s}_0) \cdot \mathbf{e}}{2(\mathbf{s}_1 - \mathbf{s}_0) \cdot (\mathbf{s}_1 - \mathbf{s}_0)}. \tag{7}$$

With $t$ now determined, the coordinates of $E'$ can be found by substituting $t$ back into the equation $\mathbf{x} = \mathbf{e} + t\mathbf{n}$, yielding $\mathbf{e}'$, the coordinates of the mapping point $E'$ on the equidistant hyperplane.

$$\mathbf{e}' = \mathbf{e} + \frac{|\mathbf{s}_1|^2 - |\mathbf{s}_0|^2 - 2(\mathbf{s}_1 - \mathbf{s}_0) \cdot \mathbf{e}}{2(\mathbf{s}_1 - \mathbf{s}_0) \cdot (\mathbf{s}_1 - \mathbf{s}_0)} \mathbf{n}, \tag{8}$$

where $\mathbf{e} \in \mathbf{E}^G$. This formula effectively yields $\mathbf{e}'$, pinpointing the location of $E'$ on the equidistant hyperplane where the line segment joining $E$ to $E'$ is perpendicular to the hyperplane, thus satisfying the geometric condition of equidistance from $E$ to points $S_0$ and $S_1$.

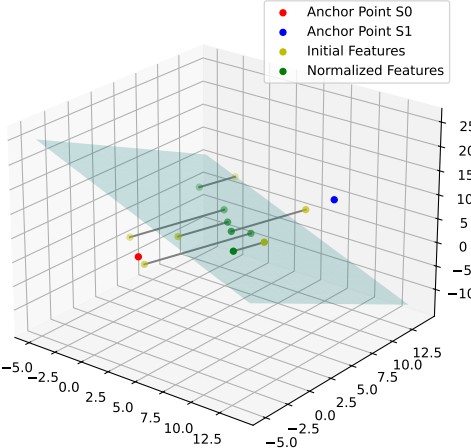

**Figure 2: Feature normalization visualization. The initial features (light green dots) of the global knowledge graph nodes are projected onto a hyperplane using our proposed feature normalization method. The normalized features (dark green dots) of the nodes are equidistant from both anchor points, ensuring unbiased representations for predicting the inclusion of nodes in the current report generation.**

## 3.2 RadGraph guided Spatio-temporal Multimodal Hierarchical Fusion

In alignment with radiologists' workflow, their diagnostic process initiates with a review of patients' previous examination data to comprehend the current state of the patient's condition. Then, they analyze the discrepancies between the current and prior examination results to discern critical information on disease progression, forming the basis for report generation. To emulate this procedural framework, we propose the integration of knowledge graphs to guide the fusion of clinical context and current chest X-rays, thereby capturing disease progression for precise diagnostic reporting.

*3.2.1 Visual Encoding.* A single chest radiological examination generates one or more chest radiographs $I = \{I_1, I_2, \cdots, I_m\}$ and a corresponding diagnostic report $R$. For a patient's current examination, we initially encode all produced images using DenseNet-121 [12], applying max pooling to ascertain the present chest radiograph features $\mathbf{I}^C \in \mathfrak{R}^{8 \times 8 \times 1024}$. Likewise, we acquire prior radiograph features $\mathbf{I}^{Pr} \in \mathfrak{R}^{8 \times 8 \times 1024}$ from the patient's former examination.

*3.2.2 Comparison of Prior and Current X-rays.* Radiologists typically examine important disease-related regions in patients' successive chest radiographs to assess disease progression, essentially comparing visual features pertinent to report content. Accordingly, we utilize the constructed knowledge graphs to extract salient visual features correlated with report narratives from both previous and current X-rays, followed by a comparative analysis of the extracted features.

Utilizing node features as queries, and relative visual features as keys and values, cross-attention mechanisms are employed to

extract node-relevant significant features from visual features.

$$E^C = CrossAtt(E'^G, I^C) \tag{9}$$

$$E^{Pr} = CrossAtt(E'^G, I^{Pr}) \tag{10}$$

$$CrossAtt(\mathbf{X}, \mathbf{Y}) \begin{cases} head_i = softmax(\frac{\mathbf{X}\mathbf{Y}^T}{\sqrt{d}})\mathbf{Y} \\ \mathbf{X}' = Concat(head_1, \cdots, head_i)\mathbf{W} \end{cases} \tag{11}$$

where $E'^G$ represents the normalized node features; $d$ is the dimension of features; $\mathbf{W}$ is a learnable weight matrix; $E^{Pr}$ and $E^C$ represent the extracted report-related key visual features of prior X-ray and current X-ray, respectively. Thus, the disease progression features, i.e., the difference of key features between current and previous X-rays can be represented as $E^I = E^C - E^{Pr}$.

*3.2.3 Spatio-temporal Multimodal Hierarchical Fusion.* To integrate the clinical context of follow-up patients, we merge knowledge graph embeddings with anchor embeddings to encode the information from prior reports, thus representing the current state features $E^S \in \mathfrak{R}^{N \times d}$ of patients. Each node in the knowledge graph is allocated a unique status determined by its occurrence in the patient's antecedent examination report: nodes reported previously are encoded with the anchor embedding $\mathbf{s}_1$ corresponding to label 1, while unreported nodes receive the anchor embedding $\mathbf{s}_0$ of label 0. For new patients, the node features are designated as null. This approach allows the initial features of the knowledge graph to effectively reflect the patient's original health condition, facilitating the integration of information from prior reports while efficiently distinguishing between follow-up and new patients.

Based on the foundational state of patients, integrating disease progression features, e.g., $E^I, E'^G$, and $E^S$, enables the current condition feature $E$ of the patient

$$E = LayerNorm(E^I + E'^G + E^S), \tag{12}$$

where $LayerNorm(\cdot)$ is from the work of [41].

To enhance the integration of inter-node dependencies, we use a self-attention mechanism to merge global contextual semantic information, thereby refining node representations.

$$E' = LayerNorm(FFN(SelfAtt(E))) \tag{13}$$

$$SelfAtt(E) = CrossAtt(E, E) \tag{14}$$

$$FFN(E) = max(0, EW_1 + \mathbf{b}_1)W_2 + \mathbf{b}_2 \tag{15}$$

However, considering the initial node features are too granular in semantic detail to compose complete textual semantics independently, we aggregate neighboring node features through Graph Attention Networks (GAT) to incorporate spatial context information within the knowledge graph. For node $i$, the updated embedding is defined as:

$$\mathbf{e}_i^{(l)} = \sigma(\sum_{j \in N_i} \alpha_{ij} \mathbf{W} \mathbf{e}_j^{(l-1)}) \tag{16}$$

$$\alpha_{ij} = softmax(att(\mathbf{W}_{e_i}, \mathbf{W}_{e_j}))$$
$$= \frac{exp(LeakyReLU(\mathbf{a}[\mathbf{W}_{e_i} \oplus \mathbf{W}_{e_j}]))}{\sum_{k \in N_i} exp(LeakyReLU(\mathbf{a}[\mathbf{W}_{e_i} \oplus \mathbf{W}_{e_k}]))}, \tag{17}$$

where $\mathbf{e}_i \in E$; $\sigma$ denotes an activation function; $N_i$ represents neighbor nodes of node $i$; $\mathbf{W} \in \mathfrak{R}^{d' \times d}$ is a learnable weight matrix; $\oplus$ is

the concatenation operation; $att$ is a feedforward neural network, parameterized by a weight vector $\mathbf{a} \in \mathfrak{R}^{2d'}$.

As illustrated, starting from the second layer, each layer is composed of a GAT, a cross-attention mechanism, and a self-attention mechanism. The second layer utilizes GAT to update the node features with one-hop neighborhood features, subsequently extracting relevant visual features under the guidance of updated node features. The third layer's update involves feature extraction utilizing two-hops neighbors, and so forth, facilitating the acquisition of visual features extracted according to varying granularities of textual semantics in different layers.

$$RradFusion \begin{cases} E'^{G(l)} = GAT(E'^{G(l-1)}) \\ E^{c(l)} = CrossAtt(E'^{G(l)}, I^C) \\ E^{p(l)} = CrossAtt(E'^{G(l)}, I^{Pr}) \\ E^{I(l)} = E^{C(l)} - E^{Pr(l)} \\ E^{(l)} = LayerNorm(E^{I(l)} + E'^{G(l)} + E^S) \\ E'^{(l)} = LayerNorm(FFN(SelfAtt(E^{(l)}))) \end{cases} \tag{18}$$

where $l \in \{2, 3, \cdots, L\}$. The output features from each layer are aggregated and subjected to layer normalization, yielding an updated knowledge graph fused spatio-temporal multimodal features.

## 3.3 Inclusive Node Prediction

The presence of each node within the report is predicted based on its distance to designated anchor nodes, thereby enabling the identification of relevant tokens within the diagnostic report.

$$E^o = LayerNorm(\sum_{L=1}^{L} E'^{(l)}) \tag{19}$$

We define the training loss according to the distances between updated node embeddings and two anchor points.

$$\mathbf{D} = [|\mathbf{e}' - \mathbf{S}_1|, |\mathbf{e}' - \mathbf{S}_0|] \tag{20}$$

$$\mathbf{P} = Softmax(\mathbf{D}) \tag{21}$$

where $S_0$ and $S_1$ represent embeddings of the anchor points, $\mathbf{e}' \in E^o$. The loss function is the cross-entropy loss

$$L_c = -\frac{1}{M} \sum_{i=1}^{M} \sum_{j=1}^{2} y_{ij}^n log(p_{ij}^n), \tag{22}$$

where $y_{ij}^n \in \{0, 1\}$ and $p_{ij}^n \in [0, 1]$ are the ground-truth label and predicted label of the $i$-th node, respectively. Updated node features are also used to predict their presence in the current report. If $p_{i1}^n$ exceeds a predefined threshold $Thred$, the $i$-th node is classified as label 1; otherwise, it is classified as label 0.

## 3.4 Report Generation

We employ a constructed global knowledge graph to guide the spatio-temporal multimodal fusion of patients' previous examinations, including chest radiographs and diagnostic reports, as well as current chest radiographs. After the multimodal fusion, we predict the nodes contained in the current report based on the integrated node features. Subsequently, these predicted nodes are utilized to generate diagnostic reports. To ensure that no critical details in the current chest X-ray images are overlooked, we integrate the

features of the X-rays and the nodes during the report generation process. The input of the decoder is defined by

$$\mathbf{H} = [\mathbf{I}^c \oplus (\mathbf{E}^o + \mathbf{E}^S)] \tag{23}$$

If the node label is 1, $\mathbf{E}^S = \mathbf{s}_1$; otherwise, $\mathbf{E}^S = \mathbf{s}_0$. Finally, we use a Transformer decoder to generate diagnostic reports.

$$R = TF - Decoder(\mathbf{H}) \tag{24}$$

The decoder is optimized with cross-entropy loss to maximize the conditional log-likelihood.

$$L_g = -\frac{1}{l} \sum_{i=1}^{l} \sum_{j=1}^{v} y_{ij} log(p_{ij}), \tag{25}$$

where $l$ is the length of generated report; $v$ represents vocabulary size; $p_{ij}$ is the probability of the $i$-th word of the report is the $j$-th word in the vocabulary; $y_{ij}$ is the corresponding ground truth. The overall loss function is defined as:

$$L = L_c + L_g \tag{26}$$

## 4 EXPERIMENT

### 4.1 Datasets

**MIMIC-CXR** [17] is a large dataset of 227,835 imaging studies involving 65,379 patients who presented to the emergency department of Beth Israel Deaconess Medical Center between 2011 and 2016. We use the official split and exclude studies without X-ray images or missing findings. **MIMIC-ABN** [31] is a subset of MIMIC-CXR proposed. Reports in MIMIC-ABN only contain abnormal sentences. We partitioned our dataset into training, validation, and testing sets following the strongest baseline, Recap [10].

### 4.2 Evaluation Metrics

To thoroughly assess the generated reports' quality, we utilize both natural language generation (NLG) metrics and factual correctness (FC) metrics. The NLG metrics we adopt include BLUE [34], ROUGE-L [23] and METEOR [2]. These metrics are designed to evaluate the descriptive accuracy of the reports by comparing them with the ground truth reports. On the other hand, FC metrics, specifically the factual-oriented metric F1-RadGraph and the clinical efficacy metric F1-Score (which leverages 14 observations from CheXbert [38]), are implemented to assess the reports' accuracy in depicting clinical abnormalities.

### 4.3 Implementation details

For our study, the PyTorch framework facilitated the model's development, which was trained on an NVIDIA Tesla V100 GPU with 32GB of memory. Input images were resized to 256x256 and processed via a pre-trained DenseNet121 to extract features, yielding a 1024x8x8 feature map. Both self-attention and cross-attention are 8-head multihead attention. A 12-layer Transformer with four attention heads was utilized for decoding. We used an Adam optimizer with a learning rate of 3e-4, 0.01 weight decay, 0.1 dropout, and a batch size of 8. Embedding dimensions were set at 256. Node prediction counts were 255 for MIMIC-ABN and 769 for MIMIC-CXR, with a node prediction threshold of 0.17. Hyperparameters were refined based on validation set performance.

## 4.4 Baselines

We conducted a comparative analysis against a wide range of state-of-the-art baselines in medical report generation. This comparison included models with a focus on cross-modal alignment such as R2Gen [7], CMN [6], and Aligntransformer [46]. We also assessed models that utilize reinforcement learning to optimize fact-related rewards, notably $M^2fact_{ENTNL}$ [30], and others like CvT-212DistilGPT2 [32] that employ fine-tuning strategies with pretrained models. Additionally, models integrating domain knowledge such as PPKED [24], M2KG [44], KiUT [13], and ORGAN [11], as well as those incorporating historical examination data like CXRmate [33] and Recap [10], were examined. Our comparative analysis also encompassed medical report generation techniques based on LLMs, specifically XrayGPT [39] and MedPaLM [40], to provide a holistic understanding of our model's standing within the current technological landscape. The metrics reported in the original papers of these models serve as reference benchmarks for our comparative analysis.

## 4.5 Main results

*4.5.1 Language quality.* As presented in Table 1, our model achieves superior performance compared to the baselines on both MIMIC-ABN, primarily involving first-visit patients, and MIMIC-CXR, which includes numerous follow-up cases. This underscores our model's capability to generate accurate reports for diverse patient types. Compared to large-scale model-based methods e.g., XrayGPT [39] and Med-PaLM [40], our model shows significant superiority in NLG metrics, surpassing Med-PaLM(562B) by 11.9% in BLEU-1 and 3% in Rouge-L on MIMIC-CXR. Relative to CXRmate [33], which also incorporates historical patient data, our approach shows enhancements of 5% in BLEU-4 and 4.3% in Rouge-L. Although Recap [10] performs best among all the baselines, its overall performance still lags behind our model, particularly in Rouge-L, where we exceed it by 1.7% and 2.5% on MIMIC-CXR and MIMIC-ABN, respectively. Despite Organ [11] and Recap [10] showing significant advantages over other baselines by integrating complex graph-building processes, this approach causes their models to be highly dependent on the quality of these constructions, reducing their generalizability.

*4.5.2 Factual correctness.* We evaluate the factual accuracy of the generated reports from different models in Table 1. Our model reaches the highest scores in both factual-oriented metric, such as F1-RadGraph(ER) and clinical efficacy metric, such as F1-Chexbert across the two datasets, cementing its preeminence in generating factually correct reports. When juxtaposed with the leading baseline $M^2fact_{ENTNL}$ [30] in terms of F1-RadGraph performance, our model exhibits a 3% improvement. While $M^2fact_{ENTNL}$ excels in F1-RadGraph, its performance is notably lower in F1-CheXbert. This discrepancy suggests that the optimization of $M^2fact_{ENTNL}$ using F1-RadGraph-related rewards may inflate its performance on this metric, potentially masking its true clinical accuracy. Compared with the best-performing Recap on F1-Chexbert, we improved 1.8% and 0.8% on MIMIC-CXR and MIMIC-ABN respectively. This advantage is particularly significant on MIMIC-CXR, which includes many follow-up patients, reflecting our model's efficacy in integrating historical patient data.

**Table 1: Comparison of NLG and FC metrics on MIMIC-ABN and MIMIC-CXR testing sets. B denotes BLEU scores; MTR denotes METEOR; R-L denotes ROUGE-L. F1-Rad denotes 1-RadGraph; F1-CE denotes F1-CheXbert; * denotes that the improvements of our model over state-of-the-art baselines on major metrics are statistically significant, based on two-tailed t-tests ($p < 0.001$).**

| Datasets | Method | NLG Metrics | | | | | | | FC Metrics | |
|---|---|---|---|---|---|---|---|---|---|---|
| | | B-1 | B-2 | B-3 | B-4 | MTR | R-L | AVG | F1-Rad | F1-CE |
| MIMIC-ABN | R2Gen [7] | 0.290 | 0.157 | 0.093 | 0.061 | 0.105 | 0.208 | 0.152 | - | 0.272 |
| | CMN [6] | 0.264 | 0.140 | 0.085 | 0.056 | 0.098 | 0.212 | 0.142 | - | 0.280 |
| | ORGAN [11] | 0.314 | 0.180 | 0.114 | 0.078 | 0.120 | 0.234 | 0.173 | - | 0.293 |
| | Recap [10] | 0.321 | 0.182 | 0.116 | 0.080 | 0.120 | 0.223 | 0.174 | - | 0.305 |
| | **Ours** | **0.322** | **0.192** | **0.125** | **0.085** | **0.128** | **0.248** | **0.183***  | **0.227** | **0.313*** |
| MIMIC-CXR | R2Gen [7] | 0.353 | 0.218 | 0.145 | 0.103 | 0.142 | 0.277 | 0.206 | 0.196 | 0.276 |
| | CMN [6] | 0.353 | 0.218 | 0.148 | 0.106 | 0.142 | 0.278 | 0.207 | 0.218 | 0.278 |
| | Aligntransformer [46] | 0.378 | 0.235 | 0.156 | 0.112 | 0.158 | 0.283 | 0.220 | - | - |
| | CvT-212DistilGPT2 [32] | 0.394 | 0.249 | 0.172 | 0.127 | 0.155 | 0.287 | 0.230 | 0.219 | 0.258 |
| | $M^2$fact$_{ENTNL}$ [30] | - | - | - | 0.083 | - | 0.269 | 0.218 | 0.320 | 0.311 |
| | XrayGPT(7B) [39] | 0.128 | 0.045 | 0.014 | 0.004 | 0.079 | 0.111 | 0.064 | - | - |
| | Med-PaLM(12B) [40] | 0.309 | - | - | 0.104 | - | 0.262 | 0.225 | 0.252 | 0.373 |
| | Med-PaLM(562B) [40] | 0.317 | - | - | 0.115 | - | 0.275 | 0.235 | 0.261 | 0.378 |
| | PPKED [24] | 0.360 | 0.224 | 0.149 | 0.106 | 0.149 | 0.284 | 0.212 | - | - |
| | M2KG [44] | 0.386 | 0.237 | 0.157 | 0.111 | - | 0.274 | 0.233 | - | 0.352 |
| | KiUT [13] | 0.393 | 0.243 | 0.159 | 0.113 | 0.160 | 0.285 | 0.225 | - | 0.321 |
| | ORGAN [11] | 0.407 | 0.256 | 0.172 | 0.123 | 0.162 | 0.293 | 0.235 | - | 0.385 |
| | CXRmate | - | - | - | 0.079 | - | 0.262 | 0.170 | 0.272 | 0.357 |
| | Recap [10] | 0.429 | 0.267 | 0.177 | 0.125 | 0.168 | 0.288 | 0.242 | - | 0.393 |
| | **Ours** | **0.436** | **0.275** | **0.184** | **0.129** | **0.177** | **0.305** | **0.251***  | **0.350***  | **0.411*** |

**Table 2: Ablation study, evaluated on validation sets.**

| Datasets | Variant | AvgNLG | F1-Rad | F1-CE |
|---|---|---|---|---|
| MIMIC-ABN | Full model | 0.181 | 0.220 | 0.317 |
| | w/o hist.info. | 0.180 | 0.217 | 0.314 |
| | w/o RadFusion | 0.174 | 0.207 | 0.294 |
| | w/o FeatureNorm | 0.173 | 0.197 | 0.276 |
| MIMIC-CXR | Full model | 0.253 | 0.351 | 0.413 |
| | w/o hist.info. | 0.248 | 0.333 | 0.401 |
| | w/o RadFusion | 0.244 | 0.328 | 0.389 |
| | w/o FeatureNorm | 0.238 | 0.314 | 0.376 |

**Table 3: The effectiveness analysis of our proposed feature normalization (FN) method on diseases with different label frequencies, evaluated on the MIMIC-ABN validation set.**

| Variant | High-freq. diseases | | | Low-freq. diseases | | |
|---|---|---|---|---|---|---|
| | P-CE | R-CE | F1-CE | P-CE | R-CE | F1-CE |
| Full model | 0.506 | 0.549 | 0.526 | 0.320 | 0.281 | 0.264 |
| w/o FN | 0.480 | 0.502 | 0.479 | 0.256 | 0.193 | 0.178 |
| Gains | +0.026 | +0.047 | +0.047 | +0.064 | +0.088 | +0.086 |

## 4.6 Ablation study

Our ablation study used validation sets to avoid overfitting the model to the testing set during hyperparameter tuning and model architecture decisions, ensuring the final evaluation is unbiased.

*4.6.1 Effect of historical information.* To examine the impact of integrating patients' historical examination data on the performance of the model, we conducted an experiment by omitting the historical information of follow-up patients. As indicated in Table 2, on MIMIC-ABN, the exclusion of historical data did not significantly affect the model's performance, which can be attributed to the fact that less than 10% of the patients on MIMIC-ABN are follow-up cases. However, on the MIMIC-CXR dataset, which contains a higher proportion of follow-up patients, the model's performance declined upon the removal of historical data, underscoring the significance of historical information for follow-up patients.

*4.6.2 Effect of RadFusion.* To validate the effectiveness of the proposed RadFusion module, we substituted it with the transformer-based Fusion module from MedKLIP [42] for integrating node and visual features. Given that the Fusion module does not account for historical information, we applied the same method of incorporating historical features as with RadFusion. As illustrated in Table 2, the model's performance deteriorated upon replacing the RadFusion module, underscoring the efficacy of the RadFusion module.

*4.6.3 Effect of Feature Normalization.* Table 2 shows that our proposed feature normalization method is the most significant among our technical innovations. To assess its effectiveness on diseases with varying frequencies, we evaluated the clinical accuracy of the generated reports in describing 14 abnormal observations of MIMIC-ABN. The diseases were categorized into two groups based on the frequency of occurrence of these observations. Those above the average frequency were considered common diseases with high frequency; Those below were classified as non-common diseases with low frequency. In Table 3, the performance improvements in low-frequency diseases are about twice as pronounced as those in high-frequency ones. This finding underscores the significance of our feature normalization method in enhancing clinical accuracy, particularly for less common diseases. This is because we mitigate the challenge posed by the long-tail distribution of tokens by addressing the frequency biases in the node feature representations.

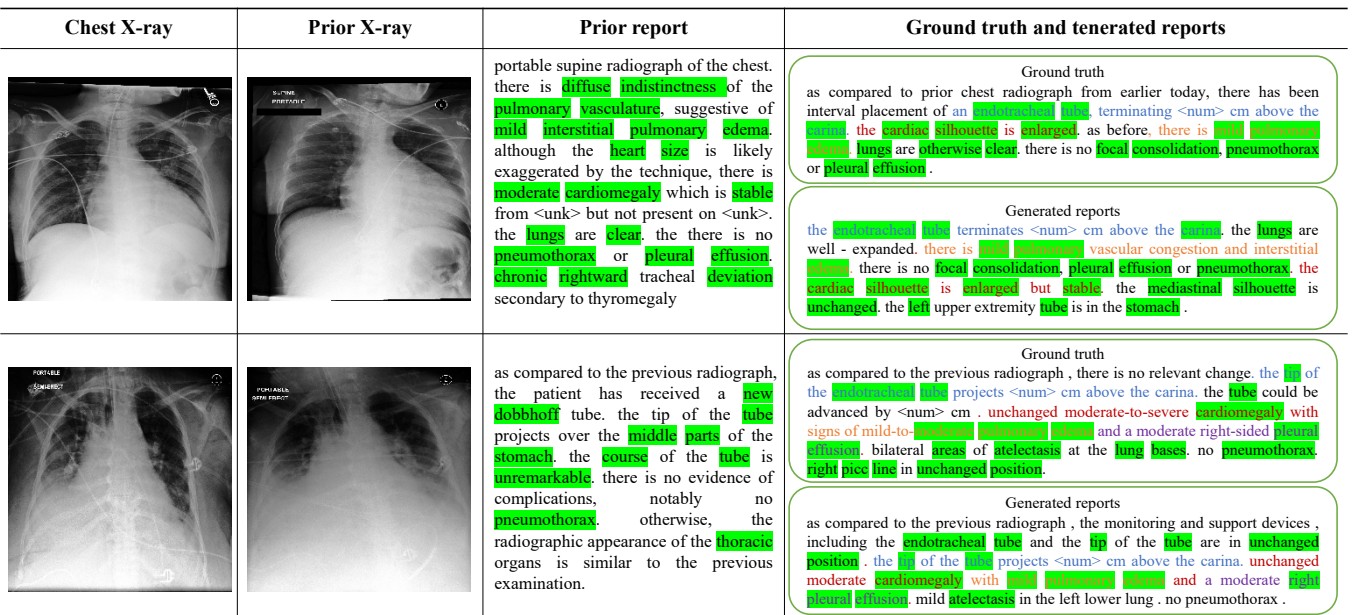

**Figure 3: Case study of two follow-up-visit samples. Predicted token nodes in reports are highlighted in green, and colored text in reports represents the abnormalities.**

## 4.7 Case Study

A qualitative analysis was conducted on two follow-up-visit samples from the MIMIC-CXR dataset in Figure 3. Our model performs a comparative analysis of patients' previous and current chest radiographs while integrating historical report information by discretizing prior reports into a series of tokens, thereby synthesizing a comprehensive diagnostic report. Within this illustration, crucial tokens predicted by our model are accentuated with green highlighting, effectively encompassing the principal semantic content of the report. Moreover, there is a significant overlap between the tokens within the generated report and the ground truth, exemplified by terms such as "endotracheal", "tube", "pneumothorax","effusion", "edema", and "enlarged". Abnormal descriptions within the report are marked in colored, with uniform coloring denoting similar disease types, underscoring our model's accuracy in delineating multiple abnormalities. For instance, in the context of support devices, the report specifies: "the endotracheal tube terminates <num> cm above the carina"; regarding cardiomegaly, it notes: "the cardiac silhouette is enlarged but stable"; and in the case of edema, it mentions: "there is mild pulmonary vascular congestion and interstitial edema." The generated reports also covers long, complex sentences describing multiple abnormalities, such as "unchanged moderate cardiomegaly with mild pulmonary edema and a moderate right pleural effusion."

## 5 CONCLUSION

In this paper, we present a medical report generation framework that emulates radiologists' workflows by integrating both historical and current patient data, enabling disease progression tracking and personalized report creation. Our method tackles textual diversity in cross-modal tasks through style-agnostic representations and advanced token prediction. We introduce a novel spatio-temporal fusion method for integrating textual and visual data across multiple granularities and apply a feature normalization technique to address biases from long-tail frequency distributions, enhancing accuracy in rare disease reporting. Experimental results on two public datasets demonstrate our model's superiority over state-of-the-art baselines.

**Limitation and Future Work.** Our enhancements include discrete node prediction to enrich reports. However, the typical scarcity of nodes in reports leads to significant data imbalances, impacting prediction accuracy. We address this by excluding some low-frequency nodes, which risks omitting critical semantic information. Future work will aim to balance node prediction tasks more effectively, ensuring the comprehensive capture of crucial diagnostic details to improve the reliability of automated medical reporting.

## 6 ETHICS STATEMENT

The datasets used in this work, MIMIC-ABN [31] and MIMIC-CXR [17], are publicly available and have been automatically de-identified to protect patient privacy. Our review confirms that the usage of these datasets poses no substantial ethical risks. However, despite the model's ability to enhance the factual accuracy of medical reports, it has not yet reached a level suitable for clinical application. The generated reports may occasionally include inaccurate or biased observations and diagnostic suggestions. We strongly recommend that healthcare professionals critically evaluate and validate the model's outputs before any clinical application to ensure patient safety and care quality.

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
