# OpenReview forum: "Medical Report Generation via Multimodal Spatio-Temporal Fusion"
_acmmm.org/ACMMM/2024/Conference — MM2024 Poster_

### Official Review · Reviewer_mJ6W · 2024-05-03

**Rating:** 4
**Confidence:** 2

**Summary:**

This article describes a medical report generation method based on multimodal spatio-temporal fusion. The method simulates the workflow of a radiologist, monitors disease progression by comparing the patient's historical and current images, and integrates previous diagnostic reports to generate a current personalized report.

**Strengths:**

1. The method proposed in the paper takes into account the patient's historical data and current images when generating medical reports, thus enabling monitoring of disease progression and generation of more personalized reports.
2. The proposed multi-granularity spatio-temporal fusion module is able to effectively fuse textual and visual features, providing significant support for report generation.
3. The introduced feature normalization method can solve the long-tail distribution problem and improve the accuracy of report generation for rare diseases.

**Limitations:**

1. The method proposed in the paper relies on a patient's past history of diagnosis and is not friendly to new patients.

2. In Figure 1, how is Global KG constructed and initialized?

3. In Figure 1, the model requires input of front and lateral view x-rays of the patient. If one of these views is missing will the model still work?

**Suitability:**

3

---

### Official Review · Reviewer_bfos · 2024-05-15

**Rating:** 4
**Confidence:** 4

**Summary:**

The article proposes a novel framework that introduces patients' past medical images and report information, compares them with current medical images, and generates new medical reports. The RadFusion module is proposed for the spatio-temporal fusion of multi-modal features. In addition, to solve the long-tail problem existing in the data set, a normalization method is proposed to ensure the robustness of rare disease report generation.

**Strengths:**

1. The problem to be solved in the article is clear, the corresponding solution is given, RadFusion is proposed and its effectiveness is demonstrated.
2. The demonstration of the ablation experiment is relatively complete.
3. The writing of the article is relatively smooth and easy to read.

**Limitations:**

1.In Section 3.1 Knowledge Graph Initialization, you mentioned that both node features and anchor point features are randomly initialized and trainable. Is this training process Inclusive Node Prediction? Can you elaborate on this training process and how the corresponding ground truth values are obtained?
2. How is b in formula (3) obtained? There is no relevant closing statement in the preceding and following paragraphs. Is it a typo?
3. There is a typographical error (e.g., should be RadFusion rather than RradFusion in Equation 18).

**Suitability:**

3

---

### Official Review · Reviewer_DFmi · 2024-05-24

**Rating:** 3
**Confidence:** 4

**Summary:**

This paper proposes a report generation framework that is consistent with the diagnostic process of radiologists, taking into account the differences between existing and past images, and generating new reports based on the style of previous reports. To address the cross-modal issues involved, the paper proposes an enhanced diagnostic report generation method that achieves the fusion of spatio-temporal information from two modalities at multiple granularities. In response to the issue of imbalanced categories of medical data, the paper also developed a feature normalization method to ensure that the prediction likelihood of each token is not affected by the frequency of occurrence. The method proposed in the paper was experimented on two public datasets.

**Strengths:**

1. Detailed description of the transformation process and mathematical process of feature normalization in feature space.
2. A novel feature extraction method considering patient's spatio-temporal information was constructed by using graphs and cross modal mechanisms, which is similar to the diagnostic process of doctors.

**Limitations:**

1. Experiments only conducted on two datasets, MIMIC-CXR and MIMIC-ABN, where MIMIC-ABN is a subset of MIMIC-CXR. More experiments on different datasets need to be conducted.

**Suitability:**

2

---

### Official Review · Reviewer_gY2q · 2024-05-24

**Rating:** 3
**Confidence:** 3

**Summary:**

In the task of medical report generation, models typically generate reports based solely on current chest X-ray images. This approach contrasts with real-world scenarios, where doctors consider historical examination data as crucial references. To address this, this paper proposes a new framework that compares past and present patient images to monitor disease progression and uses previous diagnostic reports as references to generate personalized current reports. Additionally, a novel multi-granularity spatiotemporal fusion method and feature normalization technique are introduced, effectively integrating multimodal features and ensuring unbiased token generation. Experimental results on two public datasets demonstrate that the proposed model outperforms the latest baseline models.

**Strengths:**

1. This paper identifies the challenges of automatically generating radiology reports and emphasizes that existing methods have not considered historical observation information, an issue that has not been previously addressed.
2. The proposed new framework utilizes historical observation information and integrates it using the RadFusion module and FeatureNorm, demonstrating both innovation and rationale.
3. Comprehensive experiments, including comparisons with state-of-the-art methods and ablation studies, provide evidence for the effectiveness of the proposed approach.

**Limitations:**

1. The paper does not seem to explain how the factual correctness (FC) metrics are calculated, nor does it provide relevant citations. Additionally, is the CE Metric calculated using Micro or Macro averaging?
2. Could you provide statistics on the number of follow-up cases in the dataset and conduct ablation experiments to show their proportionate impact?
3. In the ablation experiments, the contribution of historical information is noticeably less significant compared to the RadFusion module and FeatureNorm. Is it appropriate to emphasize the use of historical information as the primary contribution of the paper? Additionally, I still do not understand what the prior report should be in the absence of follow-up cases.

**Suitability:**

3

---

### Meta-Review · Area_Chair_hYF1 · 2024-07-01

**Recommendation:** Accept (Poster)
**Confidence:** 4

**Metareview:**

This paper aims to generate medical reports through Multimodal Spatio-Temporal Fusion, where past and present patient images are compared to monitor disease progression, and previous diagnostic reports are referenced to generate current ones.
The motivation demonstrates both innovation and rationale, and the proposed multi-granularity spatio-temporal fusion module is effective.
The experiments are also comprehensive.
All reviewers have expressed their concerns on the experiments, such as the insufficiency of testing dataset and the performance for new patients.
Despite this, the ACs believe that this paper offers a new perspective on medical report generation, providing inspiration for the field, and shows its effectiveness on common datasets.
Therefore, the ACs have decided to accept this paper.
The authors need to improve the paper based on the reviewer's comment, especially in further demonstrating the method's effectiveness under new datasets and the absence of historical data.